# Spanish Version of a Scale to Evaluate the Quality of Work Life in Teachers: An Adaptation of Walton’s QWL Model in the Peruvian Context

**DOI:** 10.3390/bs13120982

**Published:** 2023-11-28

**Authors:** Edgardo Muguerza-Florián, Elizabeth Emperatriz García-Salirrosas, Miluska Villar-Guevara, Israel Fernández-Mallma

**Affiliations:** 1UPG de Ciencias Empresariales, Escuela de Posgrado, Universidad Peruana Unión, Lima 15102, Peru; edgardo.muguerza@upeu.edu.pe (E.M.-F.); miluskavillar@upeu.edu.pe (M.V.-G.); 2Faculty of Management Science, Universidad Autónoma del Perú, Lima 15842, Peru; 3EP de Administración, Facultad de Ciencias Empresariales, Universidad Peruana Unión, Juliaca 21100, Peru; 4EP de Ingeniería Civil, Facultad de Ingeniería y Arquitectura, Universidad Peruana Unión, Juliaca 21100, Peru; pastorisrael@upeu.edu.pe

**Keywords:** quality of work life, dimensionality, validation, psychometric properties, Peru

## Abstract

Workers’ job satisfaction benefits the organization, which constitutes a competitive advantage. This is why the Quality of Work Life (QoWL) study has gained relevance in recent years. For this reason, various scales have been developed to evaluate this organizational variable constantly. However, to date, there is no evidence in the scientific literature of a Spanish version that evaluates the validity and reliability of these scales in a Peruvian context. Thus, this study aimed to translate into Spanish and adapt and analyze the validity and reliability of a scale to assess the quality of work life in Peruvian teachers based on Walton’s model. For this purpose, 457 regular basic education teachers from a private educational network located in the three regions of Peru were surveyed. The analyses used the Structural Equation Model (SEM) with the AMOS 24 statistical software. Confirmatory Factor Analysis provided an excellent fit model of eight factors and 31 elements (CMIN/DF = 2.351; CFI = 0.955; SRMR = 0.062; RMSEA = 0.054; Pclose = 0.052). It also demonstrated good internal consistency (α = between 0.806 and 0.938; CR = between 0.824 and 0.939; AVE = between 0.547 and 0.794). These results contribute to the study of QoWL in Peru.

## 1. Introduction

Quality of Work Life (QoWL) has been defined as the quality of the relationship between workers and the work environment, considering some human, technical, and economic factors [1]. It has been stated that this is due to the level of satisfaction of a person in their workplace, harmoniously aligned with their purpose and that of the organization, and this, in turn, with the task role that each worker performs from their place [2,3,4]. One of the purposes that initiates the greatest interest in this topic focuses on promoting positive organizational behavior that motivates workers to fulfill sustainable functions, strengthening the company’s commitment to making decisions that reflect a commitment to the well-being and diligent care for its employees by providing an optimal and favorable work environment. Although the quality of work life is a multidimensional concept [1,5,6] that has generated discrepancies among various academics, the basic idea of this model is based on the theory of Maslow’s Hierarchy of Needs [7].

He divides needs into higher and lower orders. Lower-order needs include health and safety needs, such as health benefits, protective equipment, and job security, as well as financial needs, such as wages and job benefits [8,9,10]. Higher-order needs address social needs such as time for rest and interaction with colleagues, recognition for their performance, sense of accomplishment, personal development, and aesthetic needs such as creativity. In this sense, the basic idea is that higher needs are only met when lower needs have been satisfied; we all aspire to satisfy higher needs in our daily experience. Growth forces result in movement up the hierarchy, while regressive forces push overbearing needs down the hierarchy [11].

According to Maslow [6], people feel dominated by the impulses of their unsatisfied internal needs, which determine and guide their behavior and are detailed quite well in the hierarchy of human needs. Maslow [7] contributed to a psychology of the workers’ personality, emphasizing only the internal needs of humans, considering the situation in which they act. To administer the hierarchy, it is useful to know the perception of the subordinates to form a policy that satisfies the needs of the individuals of some organization, company, and state structures. Several investigations did not scientifically confirm Maslow’s theory; some even invalidated it [12,13]. However, this theory offers a guiding scheme to answer a worker’s perception of the quality of life in their work environment since it is sufficiently well structured and is currently widely used by Human Talent Management departments in companies, educational institutions, and all kinds of workplaces.

In 1930, the concept of Quality of Work Life was introduced for the first time, and then Richard E. Walton in 1973 gave a greater description to this topic [14]. After some time, several researchers began to show their interest in individual priorities in the workplace, and other pioneers emerged, such as Hackman and Oldham in 1975 [15], Westley in 1979 [15], Nadler and Lawler in 1983 [16] and Werther and Davis in 1983 [17]. However, Walton’s model has stood out from the others for its greater consistency based on eight sub-divisions, arguing that there are several working conditions that involve the organization’s performance [18]. In the 1970s, the pioneer of QWL began to consider that the industrial community had neglected environmental and labor principles, focusing primarily on economic development and technological evolution of the time, leaving a strong need to be addressed. Walton and other researchers argued that the study of QWL, in fact, was a necessary complement to the improvement of Total Quality (TQ) [19].

Considering QoWL’s role, various research studies have analyzed its importance in educational environments. Because teaching tasks demand certain activities that occupy a large part of the time, the relationship between ergonomics in the workplace and QoWL has been investigated [20], as well as QoWL and welfare services to retain valuable teachers using these benefits as a reward the institution offers [21]. Likewise, it has been shown that job stability, salary, participatory management, rewards, and recognition play a vital role in improving the perception of QoWL [22]. On the other hand, it has been shown that the greater the integration between the teacher’s life project and the institution, the greater the QoWL. Furthermore, the same study reveals that adopting promotion and support policies for teacher retention favors the reduction of diseases [23,24].

Some studies, for example, have shown a strong link between QoWL and burnout [25], stress management [26], well-being and resignation risks [27], the organizational climate [28,29], creativity of teachers [28], and the integration of the teacher’s life project in the institution, even if they are in unfavorable working conditions [23]. Furthermore, it was evidenced that organizational health and employment status can modify the impact of QoWL [30]. It can be mentioned that the high perception of teachers’ QoWL is directly affected and integrated by the level of psychological well-being, considering happiness and sufficient job satisfaction as inclusive elements in the teaching profession [31]. It also has a strong relationship with motivation, productivity, and balance between work and non-work life [5].

Furthermore, QoWL is closely related to social sustainability in several aspects. Both elements focus on promoting assertive spaces that foster high-performance entities, the long-term well-being of people, equality and diversity, the impact on the most influential community, talent retention and corporate social responsibility. In the same way, QoWL is related to the Sustainable Development Goals (OSD) established by the United Nations General Assembly in 2015 since the promotion of a better and more equitable QoWL is essential to approaching a more just, equal, healthy and sustainable world.

The benefits of QoWL are closely linked to both employees and employers, who are involved in learning more and more about the factors that positively or negatively influence the working life of workers [32,33,34,35]. At the same time, all employees are sensitive to the changes and improvements the entity can promote in the work environment [36]. It is interesting to note the results of multiple investigations where it has been possible to demonstrate how this framework has gained popularity in research and politics. Renowned scientists have studied the predictive agents of QoWL [37,38,39,40], both in national and international contexts [1]. Faced with this, it awakens the need to know the world’s interest in learning this construct’s behavior. In response, it has been found that the ten countries that conduct the most QoWL research are India, the United States, Iran, Indonesia, Malaysia, Canada, Turkey, Saudi Arabia, Australia, and Brazil. Therefore, it is evident that there is a need to make more significant efforts for its study and implementation in South America.

On the other hand, when analyzing more than twenty definitions of QoWL focused on the worker, satisfying workers’ needs, which can be personal, psychosocial, professional, labor, and economic, stands out as a central element [41,42,43]. Seen from an approach of balance between job demands and resources, QoWL must be appropriately managed, given that its imbalance can generate demotivation, poor performance, and job dissatisfaction, in addition to stress [44].

### 1.1. The Quality of Work Life Scales (QoWL)

The importance of having valid instruments that can measure the QoWL perceived by workers in various business contexts is evident [2,45,46,47].

Walton suggests that workers are affected in their work life by dissatisfaction issues [14]. During his study, he identified that work–life responsibility can be managed from eight categories, which together form a comprehensive framework for assessing and improving an organization’s quality of work life. Consideration of these dimensions can help companies create a more satisfying and productive work environment for their employees. These dimensions or categories are human capabilities (CAP), opportunities for growth and security (OOP), adequate and fair remuneration (REM), respect for the law (RFL), social integration within the organization (SIG), social relevance of working life (SRV), safe and healthy working conditions (WKC) and work influence (WIF).

Adequate and fair remuneration (REM): It refers to the perception of receiving fair and equitable remuneration for the work performed. This includes salary, benefits, and fringe benefits [14]. Salary can decipher whether fairness in pay is being realized [19]. Studies have claimed that salary and benefits could be considered major contributors to satisfaction with quality of life at work [1]. Meanwhile, Tasdemir and Burcu [45] stated that an equitable salary is an excellent indicator for evaluating the quality of work life.

Safe and healthy working conditions (WKC): This involves having a work environment that minimizes risks to the health and safety of employees. It includes accident prevention and health promotion in the workplace [14]. Satisfying lower-order needs is a motivational driver that also helps to strengthen other important areas in the life of any worker [6]. Every employee expects the entity for which he or she works to take initiatives that focus on his or her well-being; these are truly beneficial to the employer because they have the power to help, strengthen, and nurture the workforce and make it more motivated, reliable, and satisfied [48].

Opportunities for growth and security (OOP): Employees value opportunities for personal and professional growth, such as training, skills development, promotion possibilities, conditions that provide job stability, and the creation of opportunities to use their new skills, abilities, and competencies [14,46]. Fernandes et al. [19] classified this dimension into career possibilities, salary advancement prospects, personal growth, and job security. The application of these factors ensures favorable working conditions for employees [33].

Human capabilities (CAP): The workplace should become a circle where the employee can develop their human capabilities in complete freedom [2]. To achieve this purpose, self-control has become an indispensable element in this process [14,19]. This allows employees to play an important role in the development of their skills, their performance, and their autonomy, allowing their talents to be implemented in well-designed strategies [24].

Social integration within the organization (SIG): This element refers to the way in which individuals relate and collaborate within a work environment. It involves the creation of a work environment in which employees feel part of a cohesive team and where their differences and differences are valued and respected by their leaders and co-workers. Social integration in the organization seeks to promote collaboration, effective communication, and a sense of belonging in the workplace [14,19,22].

Social relevance of working life (SRV): It is understood as the employee’s perception of the importance of the roles and tasks performed in work environments [14,19]. As long as a company promotes and carries out activities that reveal a socially responsible attitude, the employee will feel proud, predisposed, committed, and identified with the institution [32]. Abebe and Assemie [29] indicated a positive relationship with growth and development, job benefits, organizational commitment, and social relevance in working life.

Respect for the law (RFL): This element is fundamental to maintaining a fair, safe, and ethical work environment. It consists of complying with all laws, regulations, and standards applicable to the company and the industry in which it operates [14]. There are some key areas in which respect for the law is fundamental: compliance with labor laws, occupational health and safety, compliance with tax and accounting regulations, data protection and privacy, compliance with environmental regulations, and compliance with cybersecurity regulations. Compliance with the law in the workplace is important to maintain the integrity of the company, avoid legal sanctions and protect the rights of employees and other stakeholders. Employers and employees have a responsibility to know and comply with the relevant laws and regulations in their area of work [37,49].

Work influence (WIF): Employee-perceived satisfaction in the workplace extends to the non-work environment and creates positive or negative influences on the overall life of workers [1]. It also refers to a person’s ability to affect or direct opinions, decisions, or actions in a work environment. It is an important skill in the business world and can be exercised in a variety of ways. It involves the ethical and constructive use of influence to achieve shared goals and mutual benefits in the work environment, and is characterized by leadership, effective communication, knowledge and experience, conflict resolution skills, charisma, persuasion, and negotiation [14,29].

These instruments should be valid to be applied to different realities, considering that scientific evidence affirms that QoWL is economically beneficial for companies in their attempt to address a balance between work and personal life. This construct remains heavily researched [28,31] in various places worldwide, and when considering its significant organizational and academic contribution, the effect it can generate in the community is valued. Although its definition may differ from the proposal of previous studies and their measurement scales, QoWL is usually associated with job satisfaction, happiness, and well-being of the worker [6]. Below is a review of the measurement scales published in high-impact journals:

The Quality of Work Life Scale (QWL) designed by Sirgy et al. [50] has 16 items, which were evaluated using a Likert-type scale of five points (between “totally disagree” and “totally agree”). It was designed in the USA in 2001 and validated in India in 2016. The scale has three factors, which are: Factor 1, QWL with Health and Safety (QWLHS); Factor 2, QWL with Family and Pay (QWLP); and Factor 3, QWL with Knowledge (QWLK), given that each item is measured by the degree of satisfaction achieved when working in any institution. Cronbach’s Alpha was valued at 0.73, 0.62, and 0.87 for each factor, respectively [48].

The Questionnaire Quality of Work Life (QoWL) designed by Subbarayalu and Al Kuwaiti [5] presents 23 items answered using a typical Likert scale of five points (between “totally disagree” and “totally agree”). It was built in 2017 and consists of five dimensions: (i) working conditions, (ii) psychosocial factors at the workplace, (iii) opportunities for training and development, (iv) compensation and rewards, and (v) job satisfaction and job security. The scale was tested and reviewed through a Six Sigma analytical tool. Reliability tests also showed that the overall Alpha coefficient value was 0.93 for internal consistency. When testing the questionnaire using factor analysis with the varimax rotation method, the total variance explained the sum of the squared loadings as 60.31 percent.

The Quality of Work Life (QWL), designed by Beloor et al. [51] in 2019 in India, has 27 items. This scale was oriented to the textile sector and has six dimensions: (1) compensation, (2) work environment, (3) relationship and cooperation, (4) job security, (5) facilities, and (6) training and development. The instrument has a Cronbach’s Alpha of 0.875. The quantifier method is the Likert-type scale with five points ranging from (1) strongly disagree to (5) strongly agree. The instrument obtained a Cronbach’s Alpha of 0.875.

The Quality of Work Life (TQWL-42) designed by Pedroso et al. [52] scale was Validated in 2014 in Brazil; it had 143 participants. The instrument was measured with a Likert-type scale of five points (where 1 is “totally dissatisfied” and 5 is “totally satisfied”). The scale presents five dimensions: (1) biological-physical, (2) psychological-behavioral, (3) sociological-relational, (4) economic-political, and (5) environmental and organizational. Likewise, the scale is made up of 42 items, and in the CFA, it obtained a Cronbach’s Alpha of 0.85.

The Quality of Life in Work scale was designed by Marín et al. [53] in 2013 in Brazil. It had the participation of 248 workers. The instrument was measured with a Likert-type scale of five points (where 1 is “totally disagree” and 5 is “totally agree”). The scale was initially constructed following Walton’s eight factors; however, when analyzing the factor loadings of the items, only four dimensions were established, with 35 items of the 52 initially proposed. The dimensions are (1) integration, respect, and autonomy; (2) fair and adequate compensation; (3) possibilities for leisure and social life; and (4) encouragement and support. The scale obtained a Cronbach’s Alpha between 0.76 and 0.89.

The Quality of Work Life Scale (QWLS) was validated by Sinval et al. [6] in 2020 in Brazil and Portugal with a sample of 1163 multi-occupational workers. It is a 16-item self-report instrument with a Likert scale-type rating of seven points (where 1 is “very false” and 7 is “very true”). It is made up of seven factors: (1) health and safety needs, (2) economic and family needs, (3) social needs, (4) esteem needs, (5) updating needs, (6) knowledge needs and (7) aesthetic needs. For internal consistency estimates, ordinal alpha and ordinal omega were used. The results demonstrated the validity and reliability of the QWLS in both countries.

According to the background research already mentioned, there has been significant interest in developing scales to measure the Quality of Work Life construct. Previous research has seen the construct applied to the textile sector [51], economically active participants [6], higher education teachers [5], public sector managers [48], and mid-level employees in the organizational hierarchy [13]. However, existing studies on QoWL have come from countries such as India, the United States, Portugal, Brazil, China, Japan, and the United Arab Emirates. While in Peru, there is no Spanish version in the scientific literature with evidence of the validity and reliability of a QoWL scale. To fill this gap, it is necessary to carry out a study to adapt a QoWL scale for Peruvian teachers of Regular Basic Education (RBE). In this sense, validation with a 32-item QoWL scale was considered appropriate [54].

### 1.2. The Present Study

The present study aimed to adapt and analyze the validity and reliability of a scale translated into Spanish to assess the quality of work life in Peruvian teachers based on Walton’s model.

## 2. Materials and Methods

### 2.1. Sample and Procedures

The study population comprised Regular Basic Education teachers at the preschool, primary, or secondary level. The educational institutions belonging to the Adventist school network are located at the national level of the Peruvian territory. The Adventist Educational Network of Peru is a representative educational system in the region, the same one that has similar models in other South American countries. This population group has particular characteristics due to its educational philosophy and religious orientation. The academic community has taken a greater interest in their behavior and addressing topics of great impact, especially in educational environments. For this reason, it was considered pertinent to use an Adventist population for this study.

A condition to be part of the study is that the teachers had to work in an educational institution at the time of the survey. It should be noted that this study was approved by the Ethics Committee of the EPG of the Universidad Peruana Unión (2023-CE-EPG-00034). Likewise, informed consent and assent from the study institution were obtained. The study was applied during the first semester of the year 2023. A non-probability sampling was applied for convenience, and the survey was carried out through a virtual link, whose questionnaire was hosted in a Google form. The questionnaire was self-administered, and participants had to provide informed consent to administer the survey. The questionnaire was shared via email to all the teachers of the Adventist institutions, a total of 28, and a total sample of 457 teachers, who provided their answers voluntarily. See Table 1 for sociodemographic characteristics.

### 2.2. Statistical Analysis

To carry out the data analysis, two statistical software were used: (1) to evaluate the descriptive analysis of the participants’ sociodemographic data, and for the exploratory factor analysis, SPSS version 25 software was used. (2) Then, to carry out the confirmatory factor analysis and evaluate the convergent and discriminant reliability, and the adjustment of the measurement model, it was carried out using the covariance structural equation model (CB-SEM), for which it was necessary to use the software AMOS version 24. This method is highly recommended to evaluate the psychometric characteristics of measurement models [55]

### 2.3. Instrument

The eight-factor model for assessing the quality of work life proposed by Walton in 1973 [14] and adapted by Jabeen et al. [1] was used. The scale was translated into Spanish and adapted to the Peruvian context. The factors were coded as follows: human capabilities (CAP), opportunities for growth and security (OOP), adequate and fair remuneration (REM), respect for the law (RFL), social integration within the organization (SIG), social relevance of work life (SRV), safe and healthy working conditions (WKC) and work influence (WIF).

The questionnaire consisted of 32 items and was structured into three sections: the first section included the instructions for completion and the informed consent of the participants, the second section contained the items of the scale, and the third included the sociodemographic variables. To answer the items, a five-point Likert-type scale was used, ranging from 1 (“strongly disagree”) to 5 (“strongly agree”).

### 2.4. Translation Process

The back-translation method with bilingual testing was used to translate the original English version of the QoWL into Spanish. Two bilingual (Spanish-English) individuals whose first language was Spanish individually completed the translation of the QoWL from English to Spanish. A Focus Group composed of six teachers with the study’s inclusion criteria compared, discussed, and modified the translations to obtain the questionnaire’s first complete version in Spanish. Using the same approach, the Spanish questionnaire was translated back into English by two bilingual individuals whose first language was English. The English and Spanish versions of the QoWL questionnaire were tested among bilingual individuals in the target population before some final modifications were made and disseminated to the study sample.

## 3. Results

Table 2 presents the descriptive statistical results of the items, such as the mean, standard deviation, skewness and kurtosis of the scale. It is observed that the values of the skewness and kurtosis are mostly less than +/− 1.5 [56], except for items CAP2, SRV1, SRV2 and SRV3, which showed a slight non-compliance with the multivariate normality of the data. Therefore, the unweighted least squares extraction method was employed, and the maximum likelihood method was also used because it has the advantage of producing estimates that are asymptotically efficient and consistent, and with large samples, the estimate is robust to the slight violation of the multivariate non-normality assumption [57].

### Exploratory Factor Analysis

Table 3 shows the exploratory factor analysis (EFA) of the items, where it can be seen that the items are distributed in eight factors according to the variable analyzed. It is observed that there is a clear difference between the eight factors. The KMO and Bartlett test (Kaiser–Meyer–Olkin sample adequacy measure = 0.948) greater than 0.7 is high, and the Bartlett test (Sig = 0.000) is highly significant to performing factor analysis. The total variance explained in the model is 69.743%, which is greater than 50%, being human capacities (CAP) = 46.280%, respect for the law (RFL) = 7.571%, social relevance of the working life (SRV) = 3.739%, social integration within the organization (SIG) = 3.449%, safe and healthy working conditions (WKC) = 2.624%, adequate and fair remuneration (REM) = 2.217%, opportunities for growth and security (OOP) = 2.034% and work influence (WIF) = 1.829%. All the items were grouped according to their initial dimensions, except for the WIF1 item, which did not present a coefficient value in any factor and was withdrawn to carry out the following validation analyses. Then, we proceeded with the confirmatory factor analysis (CFA).

Table 4 shows the validation of the final measurement model with convergent reliability and validity. It is observed that the values of Cronbach’s Alpha (α) are between 0.806 and 0.938. These values are satisfactory since for the model to be considered at an adequate level, all the values must be above 0.70 [58]. Likewise, the composite reliability (CR) values are between 0.824 and 0.939, which is also favorable since, for it to be considered an optimal model, the values must be greater than 0.60. [59]. On the other hand, the AVE values are between 0.547 and 0.794, which is considered optimal since, with acceptable values for this indicator, they must be equal to or greater than 0.5 [60]. This means the measurement model meets all the reliability and convergent validity indicators.

Figure 1 shows the factor structure of the quality of work–life scale in the study population; in this case, they are RBE teachers.

Table 5 shows the fit indicators of the measurement model of the quality of the work–life scale. According to the results of the CFA with a structure with eight dimensions, the thirty-one items explained the eight factors (Model 1). However, not all goodness-of-fit indices were excellent, and the PClose was not estimated; therefore, the model was re-specified based on the modification index (MI) [61]. In that sense, due to the similarity in phrasing, there were correlations between the errors of some items. In this way, the measurement model was analyzed by correlating the errors as follows: e2 with e5, e14 with e15, e21 with e24, e21 with e25; e23 with e25 and e28 with e29 (Model 2), obtaining all excellent fit indices.

Table 6 presents the discriminant validity; according to Fornell and Larker [55], the measurement model is validated as long as the confidence intervals do not reach unity and the quantile covariances do not exceed the AVE.

In addition to the discriminant validation with the Fornell–Larcker criteria, we have complemented our analyses with the heterotrait-monotrait criterion to evaluate discriminant validity in this study. If the HTMT value is below 0.90, discriminant validity between two reflective constructs has been established [62] (see Table 7).

Table 8 describes the instrument in its final version, after having undergone a diligent process of content validity, EFA, and CFA, presenting reliable psychometric properties for its application.

## 4. Discussion

The objective of this study was to adapt and analyze the validity and reliability of a scale translated into Spanish to evaluate the quality of work life in Peruvian teachers based on Walton’s model [14]. This is the first study in which evidence of the validity and reliability of a QoWL scale adapted to the Peruvian context is published, taken from the Walton model (1973). However, an empirical study with a similar scale has been found in Peru without demonstrating its validity or reliability [63]. Various empirical investigations have used the QoWL scale, associating them with various variables [29,64,65,66,67,68,69] and in different contexts. However, few publications have been found in databases applied to RBE.

This scale evaluates eight dimensions distributed in 31 items; however, other scales vary in their dimensions and number of items. A quick review of them shows points of agreement in their items, although a more exhaustive analysis would be necessary to find a typical pattern between the various scales.

The factor loadings obtained a score of 0.80, while that of Jabeen et al. [1] was 0.70, and in the study carried out by Fernandes et al. [19], he scored 0.959. The composite reliability (CR) was greater than 0.70, and the average variance extracted was similar in both (AVE) > 0.50. Other validated scales have obtained similar scores [45,70].

However, the most significant difference in the present study is that the model was adjusted to the eight dimensions proposed by Walton [14] because each dimension obtained a satisfactory score (see Table 4). However, in the study carried out by Fernandes et al. [19], it was found that the model fit four dimensions when performing the Student’s *t*-test and was forced to eliminate the dimensions with the lowest factor loadings: (1) use of capabilities at work, (2) opportunities at work, (3) social integration at work, and (4) social relevance and importance at work. Both the present study and that of Fernandes [19] showed satisfactory scores in the HTMT discriminant validity tests. Consequently, the question remains whether the satisfactory results obtained with both four dimensions and the eight dimensions proposed by Walton [14] are due to some particular factor of which the author is unaware. Therefore, the possibility of continuing to study this construct in search of answers that explain this phenomenon remains open.

The findings in the present research, after examining the evidence of internal structure and reliability of the QoWL scale, are a significant contribution because it provides a duly validated instrument in an RBE context, which can be used to measure the quality of work life in Peruvian educational institutions. Although we found empirical studies that apply the QoWL scale in education [27,33,71,72], the lack of Spanish-speaking RBE publications, especially in Peru, allows this scale to be used.

Because the RBE in Peru is going through a crisis caused by political and economic factors that have generated a series of deficiencies evidenced in the low results in the Program for International Student Assessment (PISA) test, various causal factors have been listed that must do with teacher training, low salaries, and a poor-quality management system [73]. Likewise, other variables, such as job desertion, low motivation, and conflicts in internal relationships between employees, can aggravate the conditions for improving the quality of education. Although education is a fundamental right, its importance demands appropriate material and human resource management. Therefore, taking care of teachers’ QoWL is an important factor in enhancing their performance, retaining talent, and improving the quality of education.

### 4.1. Implications of the Study

Focusing on teachers’ QoWL has become a business strategy to empower and develop human talent, which will lead to the retention of successful and happier staff, making their work with students more effective. In terms of the theoretical aspect, this study contributes to the literature from a solid and diligent development of the review of the construct based on recent research. The results showed the coverage and expansion of the QoWL and its influence on the organization. Methodologically, it can be said that this study provides the scientific community with a reliable and valid measurement tool for research that aims to determine the level of QoWL in educational circles, a context scarcely studied under this topic. The adapted instrument provides simple language for a clear understanding of employees, and in turn, the adapted scale will make important contributions to studies that focus on assessing QoWL behavior.

At the business level, this study expands knowledge about the dimensions of QoWL, which can allow the top management of any organization to consider the renewal of new ways and strategies to improve employee behavior. In this sense, it becomes relevant for directors or managers to understand the factors that help workers to be happier, motivated, and committed to the institution where they provide services. This implies that QoWL will allow the managers to achieve a more optimal work environment and have better responses from workers, making QoWL an indicator of organizational growth where there is involvement from senior management to the most recent worker.

### 4.2. Limitations and Future Research

Naturally, every scientific study has certain limitations that may affect the results’ generalization. This study evaluated data collected from 28 private institutions that provide RBE in Peru, all of which are from a very different context in terms of how human talent is managed in the public sector. Future research could offer the academic community results on the behavior of this construct in comparative studies or differentiating analysis. Knowing in advance that public education in Peru has suffered certain shortcomings that this study partially reveals, in this sense, it must be taken into consideration. Future research efforts could also be focused on scrutinizing sectoral and sex differences when considering the same set of QoWL factors to obtain deeper insights. Another limitation that must be taken into account is that this research has not considered a proportionality of the participating teachers, neither in gender nor teaching level, which could generate a possible bias in the results, which is why it is suggested that future studies take into account an equitable proportion of the sample. Another limitation that must be taken into account is that this test has been evaluated only by regular basic education teachers. For future research, it is recommended to carry out validation studies with teaching and non-teaching personnel to make this study more powerful.

Furthermore, the study used samples for the author’s convenience; in this sense, future research should test the validity of the QoWL scale using representative samples to analyze its behavior. It is also necessary to consider that this research was carried out only on RBE teachers from the Adventist Educational Network, so it is impossible to generalize the results to other levels or other educational contexts. In turn, it is recommended that future empirical studies include other variables that could vary the behavior of QoWL, such as those referred to in previous studies: job satisfaction, employee retention, teaching performance, or other associated variables.

## 5. Conclusions

Today, quality of work life is a fundamental aspect of the lives of all employees, organizations and society in general. Companies that value and promote a positive work environment tend to reap significant benefits in terms of employee well-being, talent retention, talent attraction, productivity and performance, corporate image and reputation, organizational health and social impact. For this reason, the present study aimed to translate into Spanish adapt and analyze the validity and reliability of a scale that assesses the quality of work life in Peruvian teachers based on Walton’s model.

The in-depth analysis of the validity and reliability of the QoWL scale based on Walton’s model confirms the evidence found, where the AFE showed that the items of the scale are divided into eight factors with a clear distinction between them. The KMO test reached a high level (0.948 > 0.7) and Bartlett’s test reached a very significant level (Sig = 0.000). The scale also showed good internal consistency (α = between 0.806 and 0.938; CR = between 0.824 and 0.939; AVE = between 0.547 and 0.794). Similarly, the Confirmatory Factor Analysis provided an excellent fit model of eight factors and 31 items (CMIN/DF = 2351; CFI = 0.955; SRMR = 0.062; RMSEA = 0.054; Pclose = 0.052).

By developing a language-friendly QoWL measure that is only applicable to Peruvian contexts, this research has significantly advanced science by giving human resource specialists and organizational behavior researchers a trustworthy tool to aid in the formulation of various personnel management policies.

## Figures and Tables

**Figure 1 behavsci-13-00982-f001:**
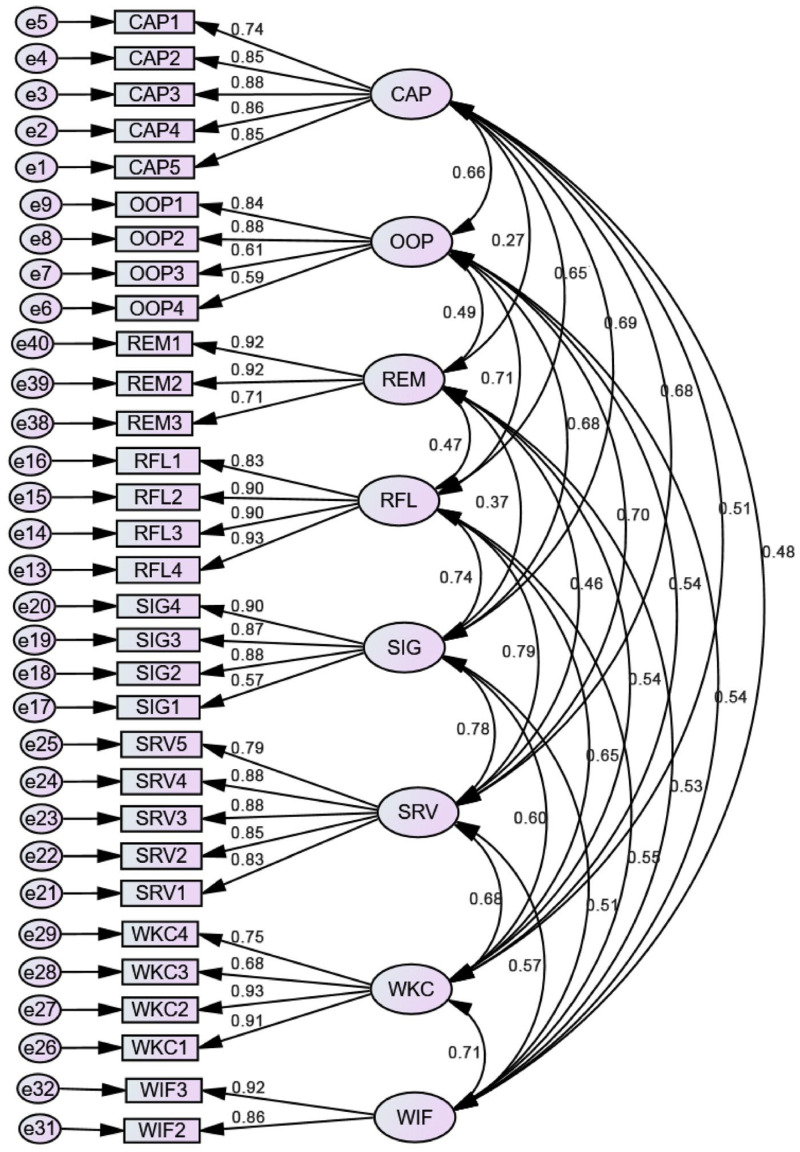
Factor structure of the quality of work–life scale.

**Table 1 behavsci-13-00982-t001:** Sociodemographic characteristics (*n* = 457).

Characteristic	Category	Frequency	Percentage (%)
Sex	Female	294	64.3
	Male	163	35.7
Age range	Up to 29 years	107	23.4
	30–39 years	151	33.0
	40–49 years	116	25.4
	50 or more years	83	18.2
Marital status	Married	271	59.3
	Single	144	31.5
	Cohabitant	18	3.9
	Divorced	17	3.7
	Widow(er)	7	1.5
Academic level	Graduated	144	31.5
	Bachelor’s degree	199	43.5
	Technician	55	12.0
	Master	58	12.7
	Doctorate	1	0.2
Teaching level	Preschool	56	12.3
	Elementary	198	43.3
	Secondary	203	44.4
Geographic location	Coast	268	58.6
	Jungle	121	26.5
	Mountain	68	14.9
Years working in the educational institution	Up to 1 year	100	21.9
	2 to 5 years	122	26.7
	6 to 10 years	65	14.2
	11 years or more	170	37.2
	Total	457	100.0

**Table 2 behavsci-13-00982-t002:** Descriptive analysis of the items (*n* = 457).

Code	Mean	Standard Deviation	Skewness	Kurtosis
CAP1	3.978	0.927	−0.936	0.921
CAP2	4.291	0.812	−1.395	2.704
CAP3	4.199	0.841	−1.100	1.432
CAP4	4.151	0.809	−0.980	1.346
CAP5	4.230	0.844	−1.247	1.838
OOP1	3.919	1.037	−0.858	0.208
OOP2	3.814	1.055	−0.839	0.279
OOP3	3.635	1.166	−0.491	−0.630
OOP4	3.007	1.343	−0.050	−1.172
REM1	2.930	1.217	−0.005	−0.910
REM2	2.921	1.208	0.047	−0.882
REM3	3.512	1.174	−0.491	−0.591
RFL1	4.059	0.936	−0.989	0.872
RFL2	4.074	0.934	−1.074	1.019
RFL3	4.116	0.875	−0.958	0.875
RFL4	4.098	0.887	−1.028	1.171
SIG1	4.101	1.117	−1.298	0.991
SIG2	4.271	0.861	−1.215	1.350
SIG3	4.070	0.929	−0.914	0.370
SIG4	4.048	0.928	−1.021	0.942
SRV1	4.249	0.835	−1.314	2.238
SRV2	4.245	0.874	−1.308	1.825
SRV3	4.239	0.862	−1.346	2.277
SRV4	4.120	0.867	−0.905	0.685
SRV5	4.026	0.973	−0.971	0.592
WKC1	3.954	0.989	−1.028	0.878
WKC2	3.926	1.019	−1.037	0.814
WKC3	3.726	1.081	−0.768	0.065
WKC4	3.777	0.984	−0.721	0.267
WIF1	2.877	1.256	−0.021	−1.001
WIF2	3.584	1.033	−0.574	−0.096
WIF3	3.746	1.001	−0.622	−0.022

**Table 3 behavsci-13-00982-t003:** Exploratory Factor Analysis (EFA) Pattern Matrix: Own elaboration.

Item	Factor							
	1	2	3	4	5	6	7	8
CAP4	0.897							
CAP2	0.878							
CAP3	0.875							
CAP5	0.739							
CAP1	0.590							
RFL4		0.908						
RFL2		0.895						
RFL1		0.821						
RFL3		0.782						
WIF1								
SRV4			0.879					
SRV3			0.864					
SRV2			0.862					
SRV1			0.693					
SRV5			0.478					
SIG3				0.851				
SIG4				0.829				
SIG2				0.815				
SIG1				0.544				
WKC2					0.953			
WKC1					0.909			
WKC3					0.608			
WKC4					0.605			
REM2						0.961		
REM1						0.929		
REM3						0.531		
OOP1							0.785	
OOP2							0.757	
OOP4							0.728	
OOP3							0.438	
WIF2								0.959
WIF3								0.716

Extraction method: unweighted least squares. Rotation method: Promax with Kaiser normalization.

**Table 4 behavsci-13-00982-t004:** Validation of the final measurement model with convergent reliability and validity.

Predictor	Items	Estimate	Alpha	CR	AVE
Human capabilities (CAP)	CAP1	0.736 ***	0.916	0.919	0.695
	CAP2	0.848 ***			
	CAP3	0.876 ***			
	CAP4	0.855 ***			
	CAP5	0.847 ***			
Opportunities for growth and security (OOP)	OOP1	0.837 ***	0.806	0.824	0.547
	OOP2	0.876 ***			
	OOP3	0.608 ***			
	OOP4	0.592 ***			
Adequate and fair remuneration (REM)	REM1	0.919 ***	0.880	0.889	0.730
	REM2	0.920 ***			
	REM3	0.706 ***			
Respect for the law (RFL)	RFL1	0.833 ***	0.938	0.939	0.794
	RFL2	0.900 ***			
	RFL3	0.900 ***			
	RFL4	0.930 ***			
Social integration within the organization (SIG)	SIG1	0.574 ***	0.868	0.887	0.668
	SIG2	0.881 ***			
	SIG3	0.866 ***			
	SIG4	0.903 ***			
Social relevance of the working life (SRV)	SRV1	0.827 ***	0.924	0.927	0.717
	SRV2	0.855 ***			
	SRV3	0.877 ***			
	SRV4	0.880 ***			
	SRV5	0.792 ***			
Safe and healthy working conditions (WKC)	WKC1	0.913 ***	0.892	0.893	0.679
	WKC2	0.931 ***			
	WKC3	0.678 ***			
	WKC4	0.747 ***			
Work influence (WIF)	WIF2	0.857 ***	0.881	0.883	0.790
	WIF3	0.920 ***			

Cronbach’s alpha (α) for all variables is >0.8, the composite reliability (CR) > 0.70, and the mean-variance extracted (AVE) > 0.50; *** *p* < 0.001 (significance level), indicating a significant validity of the model.

**Table 5 behavsci-13-00982-t005:** Statistical goodness-of-fit indices of the quality of work–life scale. Own elaboration.

Measure	Threshold	Model 1		Model 2	
		Estimate	Interpretation	Estimate	Interpretation
CMIN	-	1143.137	-	940.571	-
DF	-	406	-	400	-
CMIN/DF	Between 1 and 3	2.816	Excellent	2.351	Excellent
CFI	>0.95	0.939	Acceptable	0.955	Excellent
SRMR	<0.08	0.061	Excellent	0.062	Excellent
RMSEA	<0.06	0.063	Acceptable	0.054	Excellent
PClose	>0.05	0.000	Not Estimated	0.052	Excellent

Note: CMIN—Chi-square, DF—Degrees of freedom, SRMR—standardized root means square residual, RMSEA—Root Mean Square Error of Approximation, CFI—comparative fit index, PClose—P of Close Fit. Model 2: e2–e5; e14–e15; e21–e24; e21–e25; e23–e25; e28–e29.

**Table 6 behavsci-13-00982-t006:** Validation of the discriminant validity of the measurement model. (Fronell–Larcker criteria).

	CR	AVE	CAP	OOP	RFL	SIG	SRV	WKC	WIF	REM
**CAP**	0.921	0.701	**0.838**							
**OOP**	0.824	0.547	0.661 ***	**0.74**						
**RFL**	0.944	0.808	0.637 ***	0.696 ***	**0.899**					
**SIG**	0.887	0.668	0.692 ***	0.675 ***	0.729 ***	**0.817**				
**SRV**	0.934	0.74	0.659 ***	0.691 ***	0.771 ***	0.768 ***	**0.861**			
**WKC**	0.886	0.666	0.511 ***	0.523 ***	0.641 ***	0.588 ***	0.667 ***	**0.816**		
**WIF**	0.883	0.79	0.478 ***	0.535 ***	0.549 ***	0.506 ***	0.576 ***	0.700 ***	**0.889**	
**REM**	0.889	0.73	0.274 ***	0.486 ***	0.475 ***	0.375 ***	0.476 ***	0.527 ***	0.533 ***	**0.854**

Note: The square root of AVEs is shown diagonally in bold, *** *p* < 0.001 (significance level).

**Table 7 behavsci-13-00982-t007:** Discriminant validity of the model using the heterotroit-monotrait ratio (HTMT) criterion.

	CAP	OOP	RFL	SIG	SRV	WKC	WIF	REM
CAP								
OOP	0.661							
RFL	0.659	0.712						
SIG	0.700	0.677	0.726					
SRV	0.683	0.704	0.800	0.790				
WKC	0.528	0.629	0.68	0.622	0.724			
WIF	0.482	0.569	0.564	0.492	0.582	0.718		
REM	0.329	0.596	0.543	0.407	0.532	0.640	0.573	

**Table 8 behavsci-13-00982-t008:** Final instrument of 31 items in Spanish with an English translation.

Predictor	Measurement Ítems	Questions
Capacidades humanas (CAP) (Human progress capabilities)	CAP1	Tengo la oportunidad de tomar decisiones en el trabajo. (I have the opportunity to make decisions at work.)
	CAP2	Siento que mi trabajo contribuye significativamente al logro de los objetivos de mi organización. (I feel that my work contributes significantly to achieving my organization’s objectives.)
	CAP3	Puedo realizar diversas tareas relacionadas a mi experticia en mi centro laboral. (I can perform various tasks related to my expertise in my workplace.)
	CAP4	La evaluación de mi desempeño es satisfactoria en mi centro laboral. (The evaluation of my performance is satisfactory in my workplace.)
	CAP5	Me siento satisfecho con las responsabilidades que se me asignan en el trabajo. (I feel satisfied with the responsibilities assigned to me at work.)
Oportunidades de crecimiento y seguridad (OOP) (Opportunities for growth and security)	OOP1	Tengo la oportunidad de crecer profesionalmente en mi centro laboral. (I have the opportunity to grow professionally in my workplace.)
	OOP2	Estoy contento con la capacitación que me brindan en mi organización. (I am happy with the training they provide me in my organization.)
	OOP3	La frecuencia de renuncias en mi centro laboral es baja. (The frequency of resignations in my workplace is low.)
	OOP4	Mi organización me proporciona ayuda financiera para mi perfeccionamiento profesional. (My organization provides me with financial assistance for my professional development.)
Remuneración adecuada y justa (REM) (Adequate and fair remuneration)	REM1	Estoy contento con mi remuneración actual que la empresa me da. (I am happy with the current remuneration that the company gives me.)
	REM2	Me siento satisfecho con mi salario cuando lo comparo con el de mis compañeros. (I feel satisfied with my salary when I compare it with my colleagues.)
	REM3	Estoy satisfecho con los beneficios extras que me ofrece la empresa (bonos, viajes y otros). (I am satisfied with the company’s extra benefits (bonuses, trips, and others.)
Respeto a la Ley (RFL) (Respect for the law)	RFL1	La empresa respeta los derechos de los trabajadores. (The company respects the rights of workers.)
	RFL2	Tengo la oportunidad de expresar mis opiniones en mi centro laboral. (I have the opportunity to express my opinions in my workplace.)
	RFL3	Estoy satisfecho con las normas y las reglas de mi trabajo. (I am satisfied with the standards and rules of my work.)
	RFL4	Siento que se respeta mi individualidad en el trabajo. (I feel that my individuality is respected at work.)
Integración Social dentro de la organización (SIG) (Social integration within the organization)	SIG1	La discriminación (social, religiosa, racial, sexual, etc.) es muy baja en mi centro laboral. (Discrimination (social, religious, racial, sexual, others) is very low in my workplace.)
	SIG2	La relación con mis compañeros y jefes es satisfactoria. (The relationship with my colleagues and bosses is satisfactory.)
	SIG3	Mis compañeros y equipos están comprometidos con el trabajo asignado. (My colleagues and teams are committed to the assigned work.)
	SIG4	Mis ideas e iniciativas son valoradas por mis colegas y jefes. (My colleagues and bosses value my ideas and initiatives.)
Relevancia Social de la vida laboral (SRV) (Social relevance of the working life)	SRV1	Me siento orgulloso de trabajar en mi actual centro laboral. (I feel proud to work in my current workplace.)
	SRV2	Me siento feliz por la imagen que mi centro laboral tiene en mi comunidad. (I feel happy about the image that my workplace has in my community.)
	SRV3	Mi organización contribuye mucho a la sociedad. (My organization contributes a lot to society.)
	SRV4	Me siento satisfecho con la calidad de los proyectos realizados por mi empresa. (I feel satisfied with the quality of the projects carried out by my company.)
	SRV5	Estoy satisfecho con la manera cómo trata la empresa a sus empleados. (I am satisfied with the way the company treats its employees.)
Condiciones de trabajo seguras y saludables (WKC) (Safe and healthy working conditions)	WKC1	Estoy satisfecho con mis horas de trabajo semanal. (I am satisfied with my weekly work hours.)
	WKC2	Estoy satisfecho con mi carga laboral. (I am satisfied with my workload.)
	WKC3	Estoy satisfecho con la tecnología que la empresa me provee para trabajar. (I am satisfied with the technology that the company provides me to work with.)
	WKC4	Siento que tengo excelentes condiciones laborales en mi centro de trabajo. (I feel that I have excellent working conditions in my workplace.)
Influencia Laboral (WIF) (Work influence)	WIF2	Estoy satisfecho con la influencia del trabajo con las posibilidades de tiempo libre que tengo. (I am satisfied with the influence of work and the possibilities of my free time.)
	WIF3	Estoy satisfecho con mi trabajo y mi horario de descanso. (I am satisfied with my work and my rest schedule.)

## Data Availability

Data availability can be requested by writing to the corresponding author of this publication.

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
