# Peer review of "Spanish Version of a Scale to Evaluate the Quality of Work Life in Teachers: An Adaptation of Walton’s QWL Model in the Peruvian Context"

_behavsci, 2023, doi:10.3390/bs13120982_

Round 1

Reviewer 1 Report

Comments and Suggestions for Authors

I would like to recommend minimal changes to the authors in the following comments:

- Lines 53 - 55 - 65 - 126 - 290 -291 - 434 - 435 - 439 -  (revise citations for not being accompanied by numbering or in different reference models).

- Correct the interlinear spaces in line 68.

- Check the dates in the list of references: some are highlighted in bold and others are not 

- Revise citation line 79 [20].

- Include authorship of the proof (line 123).

- Add the authorship of the different proofs presented from line 201 onwards because it makes it difficult to differentiate and generates confusion as to whether it is the same as the one described above.  

Also recommend that they review the contents regarding: 

- The study sample is sex-biased (see line 277) by a ratio of approximately 2:1.

- Teaching level has a significant disproportion in the study sample with respect to Preschool faculty (12.3 %) and the other categories. This bias may affect the statistical results when compared with each other and the recommendation for further research (line 491).

Congratulations for such an interesting work

Author Response

Dear Reviewers,

Thank you very much for your informed comments, which helped us so much in improving the manuscript.  We appreciated the time you spent in doing this and tried our best to address all your comments.

We hope that this revised version of the paper reaches the expected standard, worthy of publication in this journal.

A detailed list of answers to your comments and suggestions is reported below.

Many thanks for your time.

Best regards,

Reviewer 2 Report

Comments and Suggestions for Authors

This Walton model was introduced in 1973 and has been widely discussed in various publications. should test the latest Quality of Work Life model. No new methods were used and no hypotheses were tested. However, in general the writing is quite adequate, but there is nothing new except for the research location

there needs to be an explanation why testing this model is only for teachers in advent schools.

Author Response

(The authors gave the same response as above.)

Reviewer 3 Report

Comments and Suggestions for Authors

This is an excellent article that provides valuable validation of an important tool.  My only hesitation is that the authors should have also distributed the survey to non-teachers to since validating an instrument like this on only one kind of sample is somewhat limiting.  This issue is partly addressed in the large sample size and the diverse sample.  Nevertheless, distributing the survey to non-teachers would have made the study more powerful since it is a validation study.  Perhaps this could be mentioned more in the discussion section.  

Comments on the Quality of English Language

Quality of English is excellent but some minor editing might be needed.  

Author Response

(The authors gave the same response as above.)
